# Challenges, Experience, and Prospects of Urban Renewal in High-Density Cities: A Review for Hong Kong

**Yidi Wang** [1], **Ying Fan** [2] **and Zan Yang** [1,3,*]

1    Tsinghua Hang Lung Center for Real Estate Studies, Institute of Real Estate Studies, Department of Construction Management, Tsinghua University, Haidian District, Beijing 100084, China

2    Department of Building and Real Estate, the Hong Kong Polytechnic University, Kowloon, Hong Kong

3    Department of Real Estate and Construction Management, KTH Royal Institute of Technology, 114 28 Stockholm, Sweden

*    Correspondence: zanyang@tsinghua.edu.cn

**Abstract:** Redevelopment in Hong Kong must be accelerated in response to urban decay and land shortages. However, due to a lack of incentives and effective policy tools under Hong Kong's floor area ratio regulations, there has been limited public–private partnerships in the urban renewal process, reducing both the public welfare and the efficiency of land use. We review the evolution of Hong Kong's density schemes for addressing urban redevelopment issues to identify the most important barriers to private sector involvement. We also summarise the international experience and identify viable policies, compare cases in Hong Kong with successful transfer of development rights (TDRs) examples, point out TDRs' shortcomings, and propose targeted policy optimisation strategies. On a practical level, this study has implications for the creation of targeted density policies to address Hong Kong's ageing infrastructure and facilitate the urban transformation of Hong Kong and similar high-density cities so that they can continue to support sustainable urban growth.

**Keywords:** floor area ratio regulation; urban renewal; transfer of development rights; density relaxation; Hong Kong

## 1. Introduction

Urban renewal is a significant issue in many nations. In recent years, urban decay, characterised by ageing buildings and a shortage of land for future urban growth, has increased the demand for urban renewal in Hong Kong. In response, the Hong Kong government has issued case-by-case approvals of land premium payment exemptions (through the Urban Renewal Authority) and density relaxations (through the Development Bureau under the Outline Zoning Plan and Building (Planning) Regulations). Nevertheless, a variety of factors are causing sluggish redevelopment momentum. The most prominent of these factors is the contradiction between developers' desire for the high returns associated with high-density development and the reality that only low-density development is possible. Due to the nature of redevelopment in large urban environments, project sites are typically located in historic city centres with high land prices and high population densities, imposing on developers the additional burden of negotiating the transfer of property rights with multiple residents. Additionally, some of Hong Kong's building blocks have plot ratios that are either equal to or higher than the maximum level permitted by existing laws and norms, further constricting developers' profit margins and decreasing the economic potential of their rehabilitation projects. Furthermore, the demands for additional public open spaces and the preservation of sites in Hong Kong's redevelopment areas require developers to design low-density projects, at the expense of profits.

From an economic perspective, Hong Kong's floor area ratio (FAR) regulations reduce private developers' profit incentives to participate in redevelopment projects, while direct fiscal subsidies increase the government's financial burden, and further increase the cost of

redevelopment. From a social perspective, some people are dissatisfied with Hong Kong's FAR regulations for urban renewal projects, as they believe they create obstacles to the implementation of redevelopment projects. From an environmental perspective, private developers have insufficient incentives to engage in environmental protection during the redevelopment process. FAR regulation has three main types, including on-site FAR bonus, land-use variance, and transfer of development rights (TDR). The policy tool of TDR proposed by the Secretary for Planning and Lands to solve these public–private problems has only been applied in cases involving heritage preservation projects and lacks practical applications in the general field of urban redevelopment.

This study identifies the obstacles to public–private redevelopment in Hong Kong's urban renewal process and discusses the optimal incentive policy based on a factual review of urban renewal experiences in Hong Kong and internationally. We focus on the following three research questions: (1) What is the core obstacle to Hong Kong's urban renewal process and what are the potential solutions to density issues under various FAR regulations? (2) Which density scheme has been the most successful internationally? (3) How can a targeted incentive policy be guaranteed to improve Hong Kong's current shortcomings?

To answer these research questions, we focus on changes to Hong Kong's FAR regulations over time and describe the latent political, legislative, and practical challenges that threaten to impede the resolution of urban redevelopment issues. We find that the private sector's unwillingness to participate in high-density projects is Hong Kong's most important urban renewal challenge.

To solve this problem, we summarise the historical development of FAR regulations at the international level. FAR regulations are imposed in urban areas to limit buildings' floor space in an effort to ameliorate adverse environmental and social problems in urban areas [1,2]. Countries such as the USA [3], Japan [4,5], India [2], China [6–8], and Singapore [9] are at different stages of their urban development and have different FAR regulations [3]. Numerous studies of FAR regulation have emphasised policy approaches, including direct limits on density, on-site density bonuses, land-use variances, and density transfer. This review summarises cases, outlines the background and progression of FAR regulatory systems, recommends future research directions, and updates practitioners on recent findings [10]. In our review of various TDR policies, we concentrate on the most advanced policies; we believe that TDR is an effective solution to FAR-related issues and can be greatly improved in Hong Kong to suit local conditions.

Finally, we distil the factors necessary for successful TDR and individually compare them with Hong Kong's current shortcomings based on our assessment of its most recent TDR case, that of the Sheng Kung Hui Compound. Thus, to ensure the feasibility and sustainability of Hong Kong's urban renewal policies, we develop optimisation strategies for redevelopment projects that are subject to FAR regulations. Specifically, we summarise the international experience to argue that the TDRs can be used as a tool to address that challenge. However, the success of TDRs depends on appropriate TDR legislation, TDR management, TDR programme design, and TDR social support. Accordingly, we also rely on an established case in Hong Kong to conduct a targeted analysis and develop suggestions for improving the practice of TDR in Hong Kong.

This paper is organised as follows: Section 2 reviews the practice of FAR regulation in Hong Kong and distils its most pressing challenge, namely the lack of private sector participation. Section 3 introduces the international development of FAR policies and practices, providing a reference for market-based policy formulation in Hong Kong. Section 4 identifies the factors that contribute to the success of FAR regulation, further verifies Hong Kong's current shortcomings by highlighting real-life cases, and proposes targeted improvement strategies. Section 5 concludes the paper.

## 2. FAR Regulation in Hong Kong's Urban Renewal Process

### 2.1. Stylised Facts of High FAR and Urban Decay

Although Hong Kong covers an area of 1106.3 square kilometres, most of the population and urban development is found on Hong Kong Island and Kowloon, which cover only 132.8 square kilometres [11]. Due to its many hills and islands, Hong Kong has very little terrain that is suitable for construction. Twenty percent of Hong Kong's land has slopes of more than 30% and is therefore undevelopable, and there is a shortage of suitable land for construction. Hong Kong's built-up area covers only 24.3% of its land [12]. In contrast to its small amount of built-up territory, Hong Kong has a large population that increased from 7,072,000 in 2011 to 7,413,000 in 2021, an increase of 4.8% in 10 years [11]. Because of this increase, Hong Kong is one of the most densely populated regions in the world.

Hong Kong's need for densely populated housing and the increasing number of ageing buildings recently accelerated urban redevelopment. According to the Monthly Digest of the Building Department, Hong Kong's Building Authority authorised the destruction of 1705 buildings since January 2005 [13]. In 2011, Hong Kong's Urban Renewal Strategy predicted that 9000 structures would be at least 50 years old by 2021 [14]. Similarly, according to Urban Renewal Authority statistics posted on 28 September 2021, by 2047, approximately 80% of the building stock in the Yau Ma Tei and Mong Kok redevelopment areas will be more than 70 years old, with more than 20% of the building stock classified as having either no or negative redevelopment potential [15]. These practical cases and official forecasts reveal the urgent need for redevelopment strategies.

An important approach to address urban decay under high FAR is urban renewal, which is essentially a process of dynamic optimization of human land systems through resource reuse and land redevelopment, with the fundamental goal of building sustainable cities [16]. It can enhance the sustainable development capacity of cities by optimizing the physical and functional space of the city, and then realize the sustainable development goals in social and economic dimensions [17].

Specifically, at the physical and functional level, urban renewal projects encourage the construction of high-quality housing [18], the rehabilitation of crumbling structures, and the efficient use of the city's land and building stock [19]. Moreover, urban renewal accompanied by energy retrofit of existing buildings presents a chance to upgrade cites' energy performance in order to increase energy effectiveness and decrease household energy cost [20,21], which allows for a higher density of use with limited resources. Further, the compact and mixed-use development around stations can increase the centralization of jobs, and generally favour public transport in cities with low density and very high growth rates that minimize sprawl [22–24]. Hence, urban renewal allows the renewal of physical space, which supports the adjustment of density and achieves a new balance of functional structures within the city [25].

At the same time, urban renewal for strategic and special spaces further promoted the sustainable development of the city at the social and economic levels [26]. On the one hand, many urban renewal projects with public attributes, including rail transit construction [27], waterfront development [28], and historic and cultural district preservation [26], have strong socioeconomic spatial effects [26] Particularly in Hong Kong, one of the goals of urban renewal is to provide adequate community facilities [14]. Hence, urban renewal can provide opportunities to address social issues and to address social integrity and social integration [29] and can promote sustainable public service attributes in cities.

On the other hand, urban renewal, which aims to create employment opportunities and enhance urban attractiveness, has caused many commercial real estate projects [30], waterfront revitalization projects [28], and other projects that effectively promote high-quality economic development and industrial transformation [31]. Redevelopment is framed by the state and much of the general populace as positive and necessary to boost economic growth [32]. In general, urban renewal can effectively improve the physical, social, and economic conditions of the city to improve the quality of life and promote sustainable urban development.

### 2.2. *The Policy Evolution of Hong Kong's Urban Renewal Process*

2.2.1. Period of Spontaneous Market Renewal (Pre–1987)

Before the 1950s, the Hong Kong government rarely intervened in the urban renewal of old districts, and its attitude towards urban renewal was basically laissez-faire. In the 1960s, the government became aware of the deterioration of the old urban districts and attempted to improve their physical environments through special initiatives. However, due to the ad hoc nature of these initiatives, along with the fact that these policies (and the implementing institutions) did not provide effective support and protection mechanisms, there were many problems with their implementation and management, resulting the urban renewal piecemeal [33,34]. The initiatives did little to improve the environment of the old districts. By the 1970s, the Hong Kong government made the construction of new towns a key element of urban development, but urban renewal at the strategic level did not receive sufficient attention [35].

During this period, urban redevelopment activities were largely market-driven, primarily advanced by private developers who actively sought out projects for demolition and construction. To achieve profitability, their main target was the redevelopment of low-density projects into high-density housing. In this period of urban redevelopment, number of developers even have become giant corporations, and almost all the older low-density areas were renewed [36], but the older high-density urban areas, which presented numerous economic and operational difficulties, were neither renewed nor improved.

2.2.2. Period of Limited Government Involvement Premised on Market Profitability (1988–2000)

Beginning in the 1980s, many buildings in Hong Kong, including public housing projects, began to deteriorate. Market-regulated mechanisms alone were inadequate to address the ageing of Hong Kong's urban structures. In 1988, the government responded by establishing the Land Development Corporation (LDC), an independent statutory department dedicated to urban renewal, to promote the regeneration of Hong Kong's older districts [35,37]. The establishment of the LDC marked the beginning of the Hong Kong government's (limited) formal involvement in urban renewal activities.

Although the LDC was a semi-private–semi-public statutory body, it did not have any statutory resumption power [38]; furthermore, it did not receive strong financial support from the government when it was established, only small loans [34]. To ensure its long-term viability, the LDC's business model focused first and foremost on its own financial balance. Accordingly, the LDC operated no differently from private developers, which sought out urban redevelopment projects with profit potential rather than addressing the problems of renewing old, high-density areas.

By the time the next phase of the Urban Redevelopment Authority was established, the LDC had undertaken 26 projects, but completed only 16, 80.5% of which were commercial buildings with the possibility of profit; only 19.5% of the projects were used for residences, public facilities, or community recreation [39]. During the 12 years of the LDC's operation, only 0.44% of the 639 hectares that the 1991 metropolitan plan designated for regeneration were renewed [38]; thus, the LDC failed to meet its original purpose of improving old districts and it was widely criticised by both citizens and the government.

Thus, it is evident that the LDC maintained the characteristics of a commercial operation, and the LDC's approach to urban renewal, with economic viability as the primary consideration, was no more successful than private developers' approach in promoting the improvement of old districts. By 2000, the LDC's renewal activities were widely questioned and criticised while the number of redevelopment projects by private developers was declining [37]. The government recognised that it must increase the level of public-sector intervention to fundamentally improve the appearance of Hong Kong's old districts and enhance the efficiency and effectiveness of urban renewal.

2.2.3. Period of Increased Government Intervention in Renewal (2001–Present)

The Urban Renewal Authority Ordinance was approved in June 2000 and the Urban Renewal Authority (URA) replaced the LDC in May 2001, with a goal of completing 225 redevelopment projects in 20 years [40]. The establishment of the URA marked the entry of Hong Kong's urban redevelopment activities into a phase led by a statutory body with governmental support. Unlike during the LDC era and the URA era, the Hong Kong government shouldered additional responsibility for urban renewal. First, the government issued the Urban Renewal Strategy to guide urban renewal activities. Second, the government increased its financial support by providing HKD 10 billion in start-up capital to the URA and offering concessions in land premium reductions. Third, the government made corresponding arrangements in the areas of planning, acquisition, and public participation [14,38,41,42].

Since the above transformation, the government increased its intervention in urban renewal in Hong Kong and became fully involved in urban regeneration activities [41]. The government has a clear plan to guide regeneration activities from top-level strategies to financial support policies, and it constantly adjusts its strategies before introducing new guidelines. However, Hong Kong's current regeneration of old districts cannot keep up with the speed of urban ageing. The current approach is insufficient to solve the problem, and its sustainability is questionable [43]. Table 1 lists the implementation rules of Hong Kong's three urban renewal periods and summarises their main features, the problems they solved, and their residual problems.

*2.3. The Challenges of Insufficient Private Sector Participation*

A variety of factors have slowed redevelopment in Hong Kong, creating an increasingly large gap between the urgent demand for and the production of new residential units. One of the most important factors is the lack of incentives for developers to participate in redevelopment projects in high-density areas [38]. At present, almost all the old, low-density and easily redevelopable districts in Hong Kong have been renewed by private developers, as these were profitable projects. However, developers' participation in urban redevelopment activities has steadily declined because of the low profitability and high economic risk of urban redevelopment in high-density areas [38].

Hong Kong's planning system includes both discretionary between development control and the respective plan, and strong legislative power on development control when making decisions on planning applications [44]. Hong Kong's legal system, with its strict restrictions on building density, has been unable to adapt to developers' preferences for high-density projects, decreasing developers' participation in urban renewal.

On the one hand, Hong Kong's FAR has reached the upper limits of development [14]. The redevelopment of old high-density areas can result in zero gain (or even less square footage than before redevelopment) if redeveloped buildings comply with the current regulations, directly leading to the low profitability of the redevelopment of old high-density areas. On the other hand, due to the nature of redevelopment, project sites are normally located in old city centres with high land prices and high population density, imposing an additional burden on developers who must negotiate with residents about the transfer price of their property rights [14]. The competing demands to create additional public open spaces and preserving old sites in redevelopment areas also require developers to design low-density products at the expense of profits [45,46]. In addition, parts of Hong Kong have an existing plot ratio equal to or even greater than the maximum permissible level under the Outline Zoning Plan and Building (Planning) Regulations [47], further limiting developers' ability to make a profit and reducing the economic potential of redevelopment projects. With respect to the government's financial support, according to the Urban Renewal Authority, HKD 20.8 billion land premium payment exemptions were provided by the government for 48 projects by May 2022 under the Urban Renewal Authority [48], indicating that the expansion of direct financial support for redevelopment incentives imposes a huge additional burden on the government's finances. Under the circumstances,

additional incentives in other forms are necessary to encourage private sector developers to participate in redevelopment projects.

**Table 1.** Evolution and comparison of Hong Kong's urban renewal strategies.

| | Period 1 (before 1987) | Period 2 (1988–2000) | Period 3 (2001–Present) |
|---|---|---|---|
| Main features | Private developer-led. | Limited government involvement, maintained a market mechanism. | The government increased its intervention and established a clear plan from top-level strategy formulation to financial support policies. |
| Problems solved | Mainly regeneration of low-density projects in old areas. | Urban renewal attracted the attention of the government which pursued targeted solutions. | The URA replaced the LDC to address the problems of inefficient urban renewal. |
| Problems remaining | No willingness on the part of private developers to participate in renewal of high-density projects. | LDC lacked a profit incentive, and urban renewal was inefficient. | There has been a lack of participation in the renewal of high-density projects necessary for urban regeneration. |
| Implementation rules | | | |
| ● Guidance Platform | None. | None. | Urban renewal strategy. |
| ● Public Accountability | None. | None. | The URA is accountable to the Legislative Council. |
| ● Financial Support | None. | Hong Kong's government provided HKD 31 million loan as start-up capital (subject to repayment). | The government injected HKD 10 billion in rolling funds. There is a right to a land premium waiver. Exemption from relevant taxes and fees. |
| ● Approval Process | None. | LDC's redevelopment projects were submitted to the government for approval on a case-by-case basis. Details of the projects were not announced. | A one-off submission by the URA to the Financial Secretary for approval. Details of the projects are announced. |
| ● Compensation | None. | Market price of '10-year-old' residential properties in the same area was the benchmark for compensation. No special rehousing compensation for tenants. | Market price of '7-year-old' residential properties in the same area as the benchmark for compensation. Special rehousing compensation for tenants. |
| ● Community Outreach and Public Engagement | Lack of consideration of social factors and public participation. | Lack of consideration of social factors and public participation. | Social impact assessment with public opposition and appeals against the development. |

## 3. International Policy Development and Recommendations

Different countries have introduced various policies to address the dilemma of FAR in urban renewal. The most important of these policies involves the lack of incentives for developers to participate in redevelopment projects in high-density areas. We examined international policies to see if they can provide inspiration for Hong Kong. Our review covered the 1960–2022 period and focused on the implementation of TDRs. We searched reliable research databases (Emerald Insight, Web of Science, ProQuest, Google Scholar, and ScienceDirect) using keywords such as 'floor area ratio', 'floor area ratio regulation', 'density regulation', 'density relaxation', 'density transfer', and 'transfer of development rights' to find research on FAR regulations. We screened out papers that were not peer-

reviewed or in languages other than English and excluded papers that did not focus on the theme of FAR regulation. Eighty appropriate publications on the development of FAR regulation were found.

### 3.1. Common FAR Regulation Practices

### 3.1.1. Traditional FAR Regulation Practices

Traditionally, a government uses direct intervention methods, such as regulatory instruments in the form of zoning, development control, acquisition, and eminent domain, along with the purchase of development rights (PDRs) programmes, to plan and supervise the FAR [49–51]. Some nations use maximum and minimum FAR regulations, in which maximum FAR regulations indirectly control the size and height of buildings and affect building density and the urban spatial structure, whereas minimum FAR regulations can be enforced to increase building density or prevent underdevelopment [4]

However, many studies have criticised the above methods for their low levels of efficiency and effectiveness [52–55], high implementation costs [53,54], and probability of triggering conflicts between the public and the private sectors [56]. Therefore, they cannot achieve the goals of regulation and optimal land-use patterns [57]. To solve these costs and problems, land-use policies have introduced market-based adjustment tools and gradually shifted from regulatory tools and comprehensive planning to voluntary and market-based planning strategies, such as public–private partnerships, infrastructure investments, and incentives [58,59].

### 3.1.2. On-Site Density Bonuses

Different countries use different incentives for FAR control, but the most common are density relaxation and density transfer. For density relaxation, density bonuses involving quid pro quo arrangements have a long history and continue to be widely practiced. In this system, the government trades additional density for funds and public facilities. For example, Boston's South Bay project increased the permissible FAR from 2.0 to 3.0. In return, the developer pledged to provide new public space, affordable housing, and HKD 1.2 million in community beautification funds (City of Boston, 2016). In Chicago, neighbourhood opportunity bonus grants provide additional development capacity in exchange for funds from developers. In Toronto, the density bonus is known as Section 37 of the Planning Act, pursuant to which the developer must provide community facilities or other benefits in return for additional height or density allowances. Market-based increases in the allowable density of future developments in exchange for developer concessions have been widely adopted by the public sector [60–64] and have incentivised investment. To encourage developers to provide public space and affordable housing, New York and Seattle have developed Incentive Zoning programmes. Seoul's National Land Planning Law states that when developers dedicate a portion of their land for public amenities, additional construction land may be permitted. Arlington, Virginia, allows developers to build at higher densities than would otherwise be allowed for projects that provide housing for low- or moderate-income households. To encourage regional revitalisation, Arlington also provides special density bonuses for specific revitalisation areas. To stimulate land assembly, Hong Kong and Singapore offer a bonus plot ratio as an incentive for developers to assemble larger urban redevelopment sites [65,66].

The findings on the utility of density bonuses and their ultimate effect on the public interest have been mixed. In addition, such bonus provisions have been severely criticised as sacrificing design quality for the sake of urban vitality, in the case of Sydney [67], or as overbuilding to take advantage of the bonus, in the case of New York City [68].

### 3.1.3. Land-Use Variance

In addition to density bonuses, land-use variances can be used to achieve density relaxation. Jou et al. (2012) investigated four case studies in Taipei and observed that

land-use codes can be flexibly changed to legalise some commercial property development to satisfy the needs of the market [69].

### 3.2. Adoption of Market-Based Instruments: Density Transfer

A density transfer allows the owner to sell unused floor area from a 'donor site' to one or more 'receiver sites' at a market- or city-determined price [70]. TDRs are the most common method of density transfer. TDRs rely on the market to compensate landowners, which encourages developers to invest in more projects [71] and balances the pressures on administrative bodies [72].

The first application of TDRs is New York's 1916 zoning ordinance [73]. Following their first use in the USA, TDR programmes spread to other Western countries, such as France [74], the Netherlands [75], Germany [76], Switzerland [77], and Italy [78], and to Eastern countries, such as mainland China [49,79], South Korea [80], and Taiwan [81]. Many countries and cities have established legislation to promote TDRs [82,83], and there is no world standard or norm. As a result, TDRs raise numerous questions, including the difficulty of assessing the value of development rights in the absence of reliable mechanisms [84] and the inefficiency of the system in residential areas [85].

We compare the advantages and disadvantages of different FAR regulations in Table 2. All three types of FAR regulations can stimulate private sector participation. However, both on-site FAR bonus and land-use variances have inherent problems that are difficult to circumvent. For example, on-site FAR tends to encourage developers to over-develop, and land-use variance is ill-suited to setting a fixed conversion pattern. Furthermore, TDRs are the target of a substantial amount of criticism. The main problem with TDRs is that designing TDR programmes is costly for local governments due to TDRs' complexity, and such programmes are unlikely to result in an efficient land allocation [86,87]. The disadvantage of TDRs is due to their temporary immaturity, resulting in imperfectly implemented programmes in different regions and controversies about their effects. In the future, together with the promotion and maturity of TDR systems, there will certainly be more opportunities for the development of TDR systems.

**Table 2.** Comparison of FAR regulations.

| | On-Site FAR Bonus | Land-Use Variance | Transfer of Development Rights (TDRs) |
|---|---|---|---|
| Maturity | Mature | Immature | Immature |
| Popularity | High | Low | Low |
| Advantages | Market-based investment incentives, with low government cost | Increased flexibility to meet market needs | Encourage developers' investments in more development and balance the pressures on administrative bodies |
| Disadvantages | Sacrifice of design quality for the sake of urban vitality, or overbuilding to take advantage of the bonus | Difficulty in changing land-use norms according to local conditions | In the absence of reliable mechanisms, assessments of the value of the right to development may create conflicts and cause poor operational efficiency |

### 3.3. The Basic Elements of TDR

#### 3.3.1. TDR Pricing

The specific form of density transfer used by governments varies. There are generally two types of pricing: (1) via a pure free-market mechanism; and (2) via a TDR 'public bank'.

With the first type of pricing, TDRs are freely traded on the market and the price is determined by supply and demand. The problem with market-based TDR pricing is that the extensive use of TDRs priced in this manner can change the land value, causing the market price for all land to fluctuate. Taipei's strategy for preventing this negative impact

is to fix the price of the development rights rather than allowing them to fluctuate with the market [83].

With the second type of pricing, the local government creates a TDR public bank that operates as an intermediate public agency in TDR exchanges. For example, King County, Washington, used general fund money and the proceeds from a dedicated portion of county property taxes to buy the TDRs to more than 90,000 acres of forested land and open space. The primary goal of TDR banks is to reduce price uncertainty and ensure stable and fair pricing [88,89]. However, some have criticised TDR banks on the ground that they distort the price determination of TDR [90–92].

### 3.3.2. Designation of Sending and Receiving Areas

The primary issue related to the location of TDRs involves the question of whether to designate a specific receiver site. There are generally two types of TDR designations: (1) dual transfer districts; and (2) single transfer districts.

Dual transfer districts usually have separate, pre-zoned sending and receiving areas. The planning agency can guide development to the areas that are the best suited to increased density. Single transfer districts allow the market to decide where transfers occur. For example, the Lake Tahoe basin and the Malibu/Santa Monica Coastal Zone have no clear spatial boundaries between sending and receiving areas. Some TDR systems allow more freedom to choose the receiving site. Taiwan's TDR enabling statute does not require planning authorities to designate specific areas as receiving areas eligible for higher-density developments. Livermore, California, allows the community to select the receiving sites [89].

### 3.3.3. TDR Transfer Ratio

The two types of TDR transfer ratios are as follows: (1) a one-to-one transfer ratio, and (2) an n-transfer ratio [93]. With a one-to-one transfer ratio, for each dwelling unit that is precluded from development at the sending site, one bonus dwelling unit is allowed at the receiving site. To create market incentives for sending area landowners and receiving area developers, many TDR programmes use an enhanced transfer ratio. In the Montgomery County, Maryland, programme, one TDR allows one bonus single-family detached residence or two multi-family units [94]. In Livermore, California, two TDRs are required for each bonus single-family residence, but only one TDR is required for two multifamily attached units. Dade County, Florida, has 18 zoning districts that are capable of receiving TDRs. In these districts, a TDR allows for various density bonuses and other exceptions from standard development requirements [95].

With an n-transfer ratio, the determination of the transfer ratio depends on the evaluation of the affected land's development potential. For example, in the Pinelands in the State of New Jersey, several factors determine the transfer ratio, including land type and location, past and present uses, and prior development history; evaluations take approximately 6 weeks [96]. The Malibu/Santa Monica programme uses acreage and slope in determining the transfer ratio; for smaller lots in old subdivisions, the programme also considers the square footage of buildable space [97].

### 3.3.4. Use of Receiving Areas

In terms of the use of receiving areas, some programmes allow the conversion of land use. For example, in Warwick Township, Pennsylvania, TDRs are granted to sending area landowners for farmland preservation, but they are used by receiving-site developers to achieve greater lot coverage within industrial zones [98]. In Burbank, California, the Media District TDR programme allows conversions from one land use to another if the reduction in vehicular trip generation achieved at the sending site equals the increase in trip generation created by the bonus development at the receiving site [99].

### 3.3.5. Other Incentives

There are also other incentives for developers to participate in redevelopment projects, such as the provision of additional development volume. In Pacifica, California, developers using TDRs can receive exemptions from open space, setback, coverage, landscaping, and parking requirements [99]. In the Pitkin County programme in Colorado, TDRs granted for the preservation of sending area land are used by receiving area developers to achieve bonus residential floor area [93]. It is also possible to obtain a development license. For example, Morgan Hill, California, provides priority to building permits for developments that include TDRs [100]. The Tahoe Regional Planning Agency, which covers a region in California and Nevada, allows landowners to create an 'allocation' by removing non-conforming structures from a sensitive stream environmental zone.

Table 3 illustrates and compares density policies in various countries/cities in terms of their general implications, partial implications and exceptions. However, only a few countries and cities have successful TDR programmes, a situation that has sparked international interest and created a need to address common, yet controversial implementation issues [101].

**Table 3.** Density policies in various countries/cities.

| | On-Site Density Relaxations | | Transfer of Development Rights (TDRs) | | | | | Main Type |
| | | | Receiving Sites Location | | Price | | Time | |
| | On-Site FAR Bonus | Land-Use Variance | City-Determined | Developer-Determined | City-Determined | Market-Determined | FAR Reserve | |
|---|---|---|---|---|---|---|---|---|
| United States | | | | | | | | |
| New York State | ● | ● | ● | × | × | ● | ○ | (a)(b)(d)(e) |
| Washington, DC | ● | - | ● | × | × | ● | ● | (a)(d)(e) |
| Washington State | ● | - | ● | × | × | ● | ● | (c)(e) |
| Nevada | ● | - | × | ● | ○ | ● | - | (b)(c) |
| Los Angeles, California | ● | - | × | ● | × | ● | - | (b) |
| California | ● | ○ | ○ | ○ | ○ | ○ | - | (a)(c)(e) |
| Florida | ● | ○ | ○ | ○ | × | ● | ○ | (c)(d)(e) |
| Maryland | ● | - | ● | × | ● | × | - | (c) |
| Canada | ● | ○ | ● | × | × | ● | - | (a)(e) |
| Australia | ● | ○ | ● | × | × | ● | - | (a) |
| Mumbai | ● | ○ | ● | × | × | ● | - | (d) |
| Singapore | ● | ● | × | × | × | × | × | × |
| Japan | ● | × | ● | × | × | ● | - | (a)(f) |
| Taiwan, China | | | | | | | | |
| Taichung | ● | × | ● | × | × | ● | - | (a)(e) |
| Taipei | ● | ● | × | ● | ● | × | - | (a)(e) |
| China | | | | | | | | |
| Guangzhou | ● | × | ● | × | × | × | - | (a) |
| Chongqing | × | × | ● | × | ○ | ○ | - | (c) |
| Hong Kong | ● | × | ● | × | × | × | × | (a) |

Notes: ●: general rules; ○: partially implied rules; ×: not applicable; -: uncertain. (a) Heritage protection; (b) environmental protection; (c) farmland protection; (d) affordable housing; (e) urban public facility; (f) transportation facilities.

### 3.3.6. Types of TDR

The objectives of TDR differ from programme to programme. Based on the different objectives of TDR, we divide TDR into six main types, which are (a) heritage protection; (b) environmental protection; (c) farmland protection; (d) affordable housing; (e) urban public facility; (f) and transportation facilities.

Traditional TDR programmes have been implemented to protect urban characteristics, as well as to preserve environmental and agricultural [89], namely types (a)(b) and (c). Historic preservation TDR programmes (a) originally emerged in large cities, including New York, and Washington, DC. Later, medium-sized cities, such as California, and some small cities have applied TDR to preserve historic sites [99]. Similar to heritage protection are environmental protection (b) and farmland protection (c). They are all transfer of development potential from nature reserves or rural to urban areas, focusing

on the protection of agricultural and environmentally sensitive lands, including wetlands, slopes, forests, natural landscapes, animal habitats, and open spaces [95].

The focus of these types of TDR is on protecting the sending area, rather than developing the receiving area. Moreover, compact development along smart growth principles is a common secondary goal or co-benefit. These TDR implementation sites tend to be in more developed cities. They have a mature urbanization stage, a strong welfare conscious government, and urban renewal policies that take into account diverse social, environmental, and cultural demands [102,103].

There are also some innovative types of TDR that are emerging, including Affordable housing (d), Urban public facility (e), and Transportation facilities (f). These programmes place greater emphasis on incentivizing development because they set their sending areas to urban buildings or facilities where private sector participation is urgently needed, while designating receiving areas in marginal areas where the need for new development is higher.

In affordable housing projects, the government is often eager to enhance the quality of life of people by improving housing [104]. Therefore, private developers receive incentives in the form of TDRs that allow additional housing construction in other parts of the city [105]. Such projects are focused in developing countries with immature urbanization and are in the initial stage of physical renewal [106], such as Mumbai [107], to improve the well-being of residents and enhance social satisfaction. There are also some such projects in countries where privatization is prevalent, represented by the United States, which focus on embedding the private sector in urban regeneration to drive urban tax revenue and employment [102].

TDR projects of Urban public facility and Transportation facilities are derived from the government's public service attributes and industrial development needs. The most prominent of them are in Asian countries/regions such as Japan and Taiwan, China. They have developed rapidly during economic globalization, with distinctive state-led characteristics of urban development [108], and urban renewal revolves around the needs of industrial activities [109,110]. Meanwhile, some developed countries have gone through rich urban renewal development stages [25,111,112] and have also extensively experimented with diverse applications in the types of TDRs [103].

The implementation of FAR regulations varies from country to country and city to city. In general, on-site FAR bonuses are the most common, whereas land-use variances are relatively rare. Most notably, TDRs have increased flexibility and effectiveness in different settings depending on whether the receiving site's location is city- or developer-determined, whether its price is city- or market-determined, and the programme's FAR reserve. Currently, the most commonly used form of TDR is a combination of city-determined receiving locations and market-determined prices. Cities with long-standing policies and a high level of policy maturity, such as New York and Washington, DC, have provided a higher degree of freedom by allowing FAR reserves as part of their TDR policies. Cities in China that have only been implementing the policy for a short period of time (or they have just started to pilot it) are more restrictive, providing the government the power to decide the receiving site's location and price and to liberalise FAR reserves. With the accumulation of practical experience in cities, there is an opportunity to implement TDRs that are both flexible and market-oriented. Hong Kong can use this flexibility to adjust its TDR policies to achieve a high level of efficiency.

## 4. Policy Outlook and Recommendations to Meet the Challenges of Insufficient Private Sector Participation

### 4.1. Adaptability of TDR in Hong Kong

TDRs are an efficient solution to the inefficiencies of urban renewal in Hong Kong and can provide developers with an effective incentive to participate in the urban renewal of high-density projects. The Hong Kong government has explored TDRs as a solution to the problem of insufficient private sector participation in urban renewal. The concept of TDRs

was proposed by the Secretary for Planning and Lands in 2001 and has been successfully applied in nine heritage preservation projects [112]. However, TDRs in Hong Kong are not formalised instruments and have only been used for projects in which the sending areas are heritage-preservation sites and the receiving areas are alternative sites where additional density can be obtained through the acquisition of development rights [71,113]. In these cases, the receiving sites are contiguous, leaving the potential flexibility and feasibility of non-contiguous receiving site selection in TDRs in Hong Kong unexplored [114].

According to the literature, TDR programmes such as that used in Hong Kong incur high transaction costs, as the government is required to spend a substantial amount of time and money to find suitable receiving sites, assess the value and capacity of the land, engage in public consultation, and conduct site assembly. The programme can also affect surrounding owners' property values, leading to lengthy negotiations to assemble smaller lots into larger redevelopment sites [115].

Although Hong Kong's initial TDR programme, as proposed in 2001, followed the practice of TDRs in the USA and Canada of providing tradable permits, freely tradable TDRs have not been available in Hong Kong due to the absence of TDR certificates under the current legislation [116]. TDRs are provided on a case-by-case basis without publishable transfer procedures or preliminary permitted transfer area planning, the latter of which depends on negotiation between the government and the property owner [117]. Furthermore, no TDR market has been established [118].

In a more general sense, because the urban structure of Hong Kong is significantly different from the urban structures of cities in the USA and Canada, the feasibility of comprehensively copying their experiences remains questionable. However, concepts similar to the Hong Kong TDR certificates with unspecified development dates date back to the 1960 Letter B system, indicating the potential dimension of time flexibility in TDRs [119].

In summary, the application of TDRs in Hong Kong's redevelopment zones is not widespread despite their potential to boost the economic viability of redevelopment projects and accelerate the pace of urban transformation. In the future, the Urban Renewal Authority can recommend TDRs as a planning tool for common redevelopment projects other than heritage preservation projects to permit the transfer of development rights from sites with extremely limited redevelopment potential to sites where expansion or increased intensity is anticipated. The key to achieving rapid urban renewal is learning how to create an appropriate and efficient TDR market that also considers Hong Kong's unique market conditions.

### 4.2. Key to the Successful Use of TDRs in Hong Kong

Research on the evaluation of density transfer programmes has been relatively limited and has primarily consisted of cases studies that analyse and summarise the factors that contribute to success [74,82,114,116,120]. Some of the more authoritative studies include Machemer and Kaplowitz, who developed an evaluative framework consisting of 13 elements found in 14 TDR programmes, including the political foundation, a consistent regulatory process, a sense of place, resources in the area that are seen as valuable, a rapidly growing area, public acceptance, appropriate receiving areas, TDR leadership, mandatory programmes, TDR banks, a TDR programme that is compatible with PDR, simplicity and cost efficiency, and knowledge of development, local land use demands, and patterns [80]. Ostrom further summarised these elements into five criteria: economic efficiency, social equity, adaptability and resilience, accountability, and conformity with general morality [121]. Pruetz and Standridge divided the common traits of TDR success factors into the three major aspects of sending area success factors, receiving area success factors, and incentive success factors [93].

Our review of the literature identified the three commonly mentioned factors: institutional and regulatory issues, TDR programme design, and social support. The specific categories and references are shown in Table 4. Institutional and regulatory issues are the foundation that anchors and sustains TDR and include TDR legislation and management.

TDR programme design requires TDR programmes first to be simple enough for developers and owners to understand and easy for government personnel to manage and operate. TDR programmes must also have sufficient incentives to attract developers, such as low transaction and management costs, receiving areas with the maximum development potential and economic incentives for participation. Because it can be challenging to implement a project without social support, TDR projects need to ensure the timely and transparent disclosure of information to the public to obtain support under effective social scrutiny.

**Table 4.** Key factors in TDRs' successes and shortcomings in Hong Kong.

| Factors | Standards | References | Hong Kong's Shortcomings |
|---|---|---|---|
| Institutional and Regulatory Issues | | | |
| TDR Legislation | TDR is anchored and sustained through a strong policy and political foundation | [80,82,93,114,116,120–125] | Lack of systematic legislation and norms; case-by-case application |
| TDR Management | Smooth TDR implementation through strong leadership and clear assignment of responsibilities | [74,80,82,93,116,120,121,123–125] | Unclear authority between departments and low management efficiency |
| TDR Programme Design | | | |
| Simplicity | Projects are easy for developers and owners to understand; projects are easy for government personnel to manage and operate | [80,82,93,114,116,120,121,123–125] | Case-by-case application, low efficiency |
| Incentives | Attracts developers through market mechanisms such as low transaction and management costs, maximum development potential for receiving areas and economic incentives for operations | [74,80,82,93,116,120–122,124,125] | Possible over-incentives |
| Social Support | | | |
| Public Support | Timely and transparent information disclosure; community monitoring mechanism | [80,82,93,114,116,120,121,123–125] | Lack of openness and accuracy of information |

We analysed a recent TDR case in Hong Kong to uncover its shortcomings in terms of the factors set forth above and determine how a TDR programme can better match Hong Kong's characteristics. The case was that of Sheng Kung Hui Compound, an important religious landmark in Hong Kong. To reduce the landmark's overall density, the government agreed to transfer 11,000 square meters of its floor area to another piece of land. However, many conflicts were revealed during the TDR project application and implementation process. These included a lack of prior consultation about the project, which led to dissatisfaction among residents due to noise and traffic; the developer's overly strong focus on obtaining private profit; the vague information provided by the TDR programme; and the unclear responsibilities of the various departments involved.

In the Sheng Kung Hui Compound case, the institutional and regulatory issues involved opposition arising from the lack of authoritative regulation of the receiving areas and their FAR ceilings. To make a case for the use of TDRs, it is critical to explain to a difficult-to-convince public that the additional FAR in the receiving area will not exceed its environmental carrying capacity. Moreover, it is difficult for all the relevant departments to agree on the communication, recognition, and cooperation needed for TDR management procedures, resulting in public confusion and questioning of the legitimacy and reasonableness of the TDR programme. The TDR programme design issues in the Sheng Kung Hui Compound case were caused by the lack of clear regulations and standards governing Hong

Kong's TDR programme, which uses a case-by-case approach. This approach is inconsistent with the standard of simplicity and ease of operation and can incur significant transaction costs. Furthermore, the TDRs used for the Sheng Kung Hui Compound case exceeded the usual incentives for commercial developers, defeating the programme's original purpose of using such incentives to encourage developer participation. The social support issues raised by the Sheng Kung Hui Compound case were evident: the case project met with strong public opposition. The public's lack of access to timely, transparent, and accurate information about the project and the lack of clear channels to participate in monitoring it undermined the principle of social equity.

*4.3. Optimisation Strategies for TDRs in Hong Kong*

Hong Kong currently uses TDRs only to a limited extent to manage historic preservation, and if it is extended to more universal urban renewal projects, the sectors and stakeholders involved are more complex. Furthermore, the projects may involve other new issues, such as challenges to the government's right to control, damage to the natural environment, and undesirable social conflicts. Therefore, we propose optimal strategies for TDRs in Hong Kong in response to the shortcomings reflected in current practice.

First, good institutional and regulatory support must be established. Legislation is the key to controlling urban planning and land development and utilisation. The government should take sustainable urban development as its guiding principle and continuously establish and improve a feasible TDR system that includes implementation processes, steps and methods, all supported by a strong enforcement mechanism. In addition, the TDR management agency should be empowered to clarify its responsibilities and powers, thus facilitating its future communication and coordination with various departments and reducing transaction costs.

Second, a scientific TDR programme design is necessary to make the TDR programme clear and easy to understand. The implementation of new TDR projects must be accompanied by reasonable and science-based TDR pricing, designation of sending and receiving areas, transfer ratios, usage of receiving areas, and other incentive policies.

Third, a higher level of social support is required. It is necessary to establish an information disclosure mechanism for the TDR programme. In addition, the public should be made aware of its role in TDRs. This can be achieved by educating the public about TDRs, publicising the programme's benefits, monitoring the development and construction of receiving areas and sending areas, and maintaining fairness to encourage public monitoring of TDR projects.

## 5. Discussion and Conclusions

### 5.1. Discussion of the Research Questions

#### 5.1.1. Outputs of the Three Research Questions

We proposed three research questions: (1) What is the core obstacle in Hong Kong's urban renewal process and potential solution of density schemes under various FAR regulations? (2) Which density scheme outperforms based on successful precedents in the international arena? (3) How to ensure the incentive policy targeted to improve the current deficiencies in Hong Kong?

We completed in-depth research and argumentation on three research questions. For the first question, we sorted out the historical policy evolution of Hong Kong and argue that at this stage, the most important obstacle in the process of urban renewal in Hong Kong is the insufficient participation of the private sector. There is an urgent need to stimulate the participation of private developers in the urban renewal process through certain policies. Through literature review, we found that the solution to this obstacle is the stimulation of density schemes, including on-site density bonuses, land-use variance, and density transfer. This answer captures the fundamental contradiction for the complex proposition of urban renewal and provides the set of policies to be used to solve it. For the second question, we compared the maturity, advantages, disadvantages, and room

for improvement of various density scenarios. The answer concludes that TDR has better results and stronger application promotion value at present. This answer is a very relevant choice of a cutting-edge option in the policy set, and points to a new urban renewal path for Hong Kong. For the third question, we sorted out the success factors of the selected incentive policies and evaluated the existing cases in Hong Kong accordingly. We answered three key shortcomings of Hong Kong's current TDR approach, namely the lack of a well-developed system and regulation, the absence of a design system for TDR, and the low level of community participation and support. Their targeted solutions are to accelerate the legislative and regulatory system, to emphasize simplicity and science in TDR design, and to increase the public transparency of TDR information. This answer provides concrete guidelines for implementing TDR policies on the ground in Hong Kong.

5.1.2. Future Research Outlook

Further research can carry out more targeted localization research on the success factors and key performance indicators of TDR policy in Hong Kong by in-person interviews with different stakeholders. As for success factors, the purpose of the interview is to gather opinions from representatives of the government (planning department, buildings department, and lands department), business sectors (developers in the residential and commercial real estate industry with or without experiences in certain districts), and academia (experts in the fields of urban economics, urban planning, and housing studies) in order to determine the relative importance of the various factors. As for key performance indicators, stakeholders include members of industry sectors (professionals in the residential and commercial real estate industry) and members of the general public (households living in the ageing building and their neighbourhoods). Hence, the further research based on the assessment from stakeholders should identify a set of success factors in FAR regulations as precautionary measures to ensure the efficiency and effectiveness of policy implementation, and develop the performance indicators to help monitor and control the delivery of redevelopment project.

*5.2. Conclusions*

We explored feasible options and optimisation strategies for urban renewal policies in high-density cities such as Hong Kong. We first reviewed the evolution of Hong Kong's urban renewal policies and identified increasing government involvement in urban renewal. The participation of the government in urban renewal has alleviated some of the difficulties associated with promoting urban renewal in a fully market-oriented environment, specifically the low returns associated with the renewal of old high-density areas. However, it has also imposed a considerable financial burden on the government and has not overcome the problem of weak private sector participation. Therefore, Hong Kong should introduce more incentive-based policy instruments to accelerate the urban renewal process in high-density areas by encouraging public–private partnerships.

To this end, we conducted a literature review of studies of FAR regulation that spoke to a policy solution for Hong Kong. We found that FAR regulations tended to transform over time from direct regulation to incentive policies. The core issue of FAR regulation is the need to address developers' unwillingness to participate. This unwillingness is attributable to the contradiction between developers' desire for profitable high-density development and cities' need for low-density planning to improve quality of life. Direct regulation does not increase developers' willingness to participate, and thus slows down the process of urban renewal. In contrast, incentive policies can attract the participation of the non-government sector through a market-based approach. TDRs, a new instrument available under the current FAR regulations, permits development rights to be moved from one zone to another. The application of TDR has been effective in many countries and regions because they have established comprehensive policies and systems to establish scientifically based pricing to designate sending and receiving areas, to define the transfer ratio, and to choose receiving areas according to local conditions.

The literature has fully affirmed the effectiveness of TDRs in stimulating developer participation and achieving appropriate FAR. The reason for TDRs' international success lies in sound TDR legislation and regulations, a strong management body, a simple and motivating programme design, and strong social support. However, judging from the practical case in Hong Kong, all these aspects are underdeveloped. To better promote the application of TDR, we propose the following three targeted improvement measures: (1) create and enhance a workable TDR programme, together with a series of implementation procedures, actions and techniques, all backed by a powerful enforcement mechanism; (2) ensure that the TDR programme is designed scientifically and includes acceptable TDR pricing, designations of sending and receiving areas, a transfer ratio, the use of receiving areas, and other incentive programmes; and (3) increase social acceptance of TDRs by promoting the public oversight of TDR programmes.

**Author Contributions:** Conceptualization, Y.F.; methodology, Y.F. and Y.W.; formal analysis, Y.F. and Y.W.; data curation, Y.W.; writing—original draft preparation, Y.W.; writing—review and editing, Y.W. and Y.F.; supervision, Z.Y.; funding acquisition, Y.F. All authors have read and agreed to the published version of the manuscript.

**Funding:** This research was funded by the Public Policy Research Funding Scheme of The Government of the HKSAR (Tackling Ageing Buildings and Facilitating Urban Transformation: Optimization of Floor Area Ratio Regulation in Hong Kong's Urban Renewal Process).

**Institutional Review Board Statement:** Not applicable.

**Informed Consent Statement:** Not applicable.

**Data Availability Statement:** The data is available on request from the corresponding author. The data is not publicly available.

**Conflicts of Interest:** The authors declare no conflict of interest.

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
