# Peer review of "Challenges, Experience, and Prospects of Urban Renewal in High-Density Cities: A Review for Hong Kong"

_land, doi:10.3390/land11122248_

Round 1

Reviewer 1 Report (Previous Reviewer 1)

The manuscript has substantially improved to the point where it can be considered a new submission. The objective of the manuscript, which is to provide solution to increase renewal operations in Hong Kong, is now clearly stated by the authors. The manuscript is well-structured and easy to read. It first explains the evolution of Hong Kong’s densification strategies, namely FAR regulations, makes a literature review and analysis of densification strategies in other countries and proposes solutions to improve regulations regarding that matter for Hong Kong.

It is understood that urban renewal is necessary not only to renovate an aging housing stock but also to provide housing for a growing population. However, the manuscript does not explicitly mention how urban renewal, by increasing density, could contribute to the sustainable development of the city (e.g., energy retrofit, increased quality of life, mixed-use densification close to public transport avoiding excessive urban sprawl). Chapter 2.1 could include this point. Besides, Chapter 2.1 refers to some official documents. The relevant citation should be integrated in the text/references.

Typo line 106: “cWarry” 

The same comment applies to Chapters 2.2 and 2.3, where some relevant citations are missing to support the different points.

In the conclusion, it would be interesting to add a comment on the next steps of this work.

Author Response

Dear Reviewer,

Thank you very much for providing the opportunity to revise our manuscript. We have carefully revised our paper incorporating the comments from the reviewers.

All the changes in this version of manuscript are marked up using the “Track Changes” function. Please see the attachment.

Yours sincerely,

Yidi

Reviewer 2 Report (Previous Reviewer 3)

Generally this resubmission addresses my previous comments, except for further explanations required for original paper’s Table 1.

The 3 research questions are adequately supported and justified as appropriate. But a discussion of the outcomes of the three research questions is needed in the Discussion section before the Conclusion.

The three new Tables contribute useful and informative summaries. Unfortunately, the discussion requested by the reviewer for original Table 1, now Table 3, on the various types (a-f) of land-use reasons for the TDR was not provided.  This variation between the listed countries and cities requires amplification.

Finally, the original submission was extremely well-written but this is not the case in the new text. There are certain sentences that are not easy to understand. The resubmission requires a professional editor of English.

Author Response

Dear Reviewer,

Thank you very much for providing the opportunity to revise our manuscript. We have carefully revised our paper incorporating the comments from the reviewers.

All the changes in this version of manuscript are marked up using the “Track Changes” function. Please see the attachment to find a point-by-point response.

Yours sincerely,

Yidi

Round 2

Reviewer 2 Report (Previous Reviewer 3)

The resubmission has addressed the all comments this reviewer made.  The paper is now convincing and the arguments are well-supported; in particular the lines between 441-443 and 447 – 452 are very well-expressed.

Again Table 3 is very interesting and could provide a platform for further research into the various land-use issues in specific countries that are relevant to TDRs. 

Some minor corrections: -

Line 237 …includes both discretionary ‘land?’ between …

Line 433   …to preserve ‘environmentally significant land’ and agricultural ‘land’ …

Line 435 …like ‘in’ California…

Line 531 ‘summarise’

Line 662 ‘specifically’ 

Table 3 - Under United States, I suggest using * to distinguish between States and cities.

This manuscript is a resubmission of an earlier submission. The following is a list of the peer review reports and author responses from that submission.

Round 1

Reviewer 1 Report

The paper attempts to make a literature review (narrative review) of the evolution of floor area ration regulations. The subject is an important issue given the need to regenerate the aging building stock of cities and limit urban sprawl. However, the scientific coherence of the work presented in the manuscript has many flaws. Put differently, the research’s design – or the way the literature review is conducted – is not convincing in terms of its scope and parameters. It analyses, at the same time, international literature on the subject (from American, European, and Asian countries) and very precise/local examples of medium-sized cities. As a result, the examples coming from the review remain anecdotal. Chapter 4 is supposed to highlight the evaluations of FAR regulations in the literature review but remains superficial. It is especially the case in section 4.2 where it only lists evaluation criteria without revealing relevant outputs.

Finally, we find the major problem of the manuscript in the conclusion (which is not really a conclusion but part of the research). Section 5.2 explains the situation of Hong Kong regeneration’s needs versus density regulations. Only at this point does the reader understand what the interest of the authors is: suggesting a possible line of approach to evolve the density rules in Hong Kong. The findings from the literature review, which are from different contexts, cannot be transposed to Hong Kong’s context.  This pitfall is, by the way, admitted by the authors in a few places in the manuscript (4.1 line 288 to 230, 5.1 line 272 to 275, 5.2.2 lines 324 to 328). To better organize the literature review – and, by consequence, the manuscript – the research should start by explaining the objective related to Hong Kong's needs. Maybe the research needs to mature further to achieve a greater coherence.

Reviewer 2 Report

In my opinion, the article is not a scientific text. It is not clear why: 1) the authors conducted their research; 2) what impact this research has/may have on the progress of science in the world.

Reviewer 3 Report

This is a well-written and well-structured paper on an important aspect of current urban renewal.   Given the long timeframe of the planning instrument - 1960 to 2022- ie about 62 years, it would be useful to have a paragraph on the history or a summary of the main trends of FAR and TDR in certain countries.

The Table on the 'Density policies in various countries/cities' is most interesting; particularly the final column indicating the main reasons for the TDR.  Again, a paragraph amplifying the six types and their specific focus in relevant cities/countries could be most illuminating and could suggest specific aspects for further research. For example,the specificity of certain cities is partially addressed in section 2.2, eg Hong Kong has highly specific geomorphic issues in terms of urban renewal.

Due to the clarity and succinct presentation of the issues, I suggest an expanded discussion about further research in this area is called for in the Conclusion.